# Pneumococcal Competition Modulates Antibiotic Resistance in the Pre-Vaccination Era: A Modelling Study

**DOI:** 10.3390/vaccines9030265

**Published:** 2021-03-16

**Authors:** José Lourenço, Yair Daon, Andrea Gori, Uri Obolski

**Affiliations:** 1Department of Zoology, University of Oxford, Oxford OX1 3SZ, UK; 2School of Public Health, Faculty of Medicine, Tel Aviv University, Tel Aviv 69978, Israel; yair.daon@gmail.com; 3Porter School of the Environment and Earth Sciences, Faculty of Exact Sciences, Tel Aviv University, Tel Aviv 69978, Israel; 4NIHR Global Health Research Unit on Mucosal Pathogens, Division of Infection and Immunity, University College London, London WC1E 6BT, UK; a.gori@ucl.ac.uk

**Keywords:** antibiotic resistance, *Streptococcus pneumoniae*, ecological competition, immunological competition, pre-vaccination

## Abstract

The ongoing emergence of antibiotic resistant strains and high frequencies of antibiotic resistance of *Streptococcus pneumoniae* poses a major public health challenge. How and which ecological and evolutionary mechanisms maintain the coexistence of antibiotic resistant and susceptible strains remains largely an open question. We developed an individual-based, stochastic model expanding on a previous pneumococci modelling framework. We explore how between- and within-host mechanisms of competition can sustain observed levels of resistance to antibiotics in the pre-vaccination era. Our framework considers that within-host competition for co-colonization between resistant and susceptible strains can arise via pre-existing immunity (immunological competition) or intrinsic fitness differences due to resistance costs (ecological competition). We find that beyond stochasticity, population structure or movement, competition at the within-host level can explain observed resistance frequencies. We compare our simulation results to pneumococcal antibiotic resistance data in the European region using approximate Bayesian computation. Our results demonstrate that ecological competition for co-colonization can explain the variation in co-existence of resistant and susceptible pneumococci observed in the pre-vaccination era. Furthermore, we show that within-host pneumococcal competition can facilitate the maintenance of resistance in the pre-vaccination era. Accounting for these competition-related components of pneumococcal dynamics can improve our understanding of drivers for the emergence and maintenance of antibiotic resistance in pneumococci.

## 1. Introduction

The Gram-positive bacteria *Streptococcus pneumoniae* (or pneumococci) remain the most common cause of pneumonia in children and the elderly. Pneumococcal invasive infections are a major cause of death in children, and are likely responsible for over 500,000 annual deaths [1]. Most carriers of pneumococci are asymptomatic, and only a small proportion develops an invasive disease such as meningitis or bacteraemia [2]. Vaccination remains the most successful public health strategy to date of reducing pneumococcal carriage and invasive disease [3,4].

Pneumococci are characterized by a polysaccharide capsule comprising over 100 distinct antigenic types, termed serotypes [5,6]. Pneumococcal conjugate vaccines target seven (PCV7), ten (PCV10) or thirteen (PCV13) serotypes. Since vaccine introduction, a rapid decrease in pneumococcal carriage and disease of the targeted serotypes has been observed [7,8,9,10,11]. Concomitantly, increases in the frequencies of non-targeted serotypes have been recorded both in carriage and disease [12,13,14,15,16,17,18,19]. This phenomenon is known as vaccine-induced serotype replacement (VISR), whereby the removal of targeted serotypes allows non-targeted serotypes to fill their immunological and ecological niches due to the removal of cross-immunity or direct resource competition [20,21,22,23,24]. Changes in the genetic composition of pneumococci have also been observed in the post-vaccination era [25,26]. These genetic changes can reflect perturbations in competition for resources among pneumococci sharing metabolic niches, also termed vaccine-induced metabolic shifts (VIMS) [27,28,29].

A second major route of combating pneumococcal carriage indirectly is by antibiotic treatment of invasive disease [30]. Similarly to vaccination, the effectiveness of antibiotics is often impaired by emergent antibiotic-resistant strains. Generally, vaccination has had a beneficial impact on the frequencies of antibiotic resistance among pneumococci, since vaccine types (serotypes targeted by vaccination, VT) often have increased antibiotic resistance [31,32,33,34]. However, post-vaccination increases in resistance among non-VTs (NVT) have been reported [19,35,36].

Previously, we have put forward hypotheses regarding the possible drivers of post-vaccination increases in resistance among NVTs. VIMS, for example, posits that the genetic profiles associated with resistance in VTs can shift to NVTs enjoying relaxed metabolic competition pressures in the post-vaccination era [28]. Furthermore, in a study by Obolski et al. [37], we have shown that vaccination can reduce within-host competition between co-colonizing serotypes. This reduction in competition may benefit antibiotic resistant strains disproportionately by relaxing the fitness cost of resistance. Vaccination can thus ameliorate the antibiotic resistance cost for NVTs and facilitate the increase in frequency of antibiotic-resistant NVTs, as has been observed [19,35,36].

In this study, we study the co-existence of pneumococcal susceptible and resistant strains in the pre-vaccination era. We do so by generalizing the original framework into an individual-based model. Such a generalization allows us to explore the theoretical effects of stochasticity, host population structure and mobility. We perform a series of sensitivity analyses based on the equilibrium states of stochastic simulations of the pre-vaccination era dynamics. We explore how two within-host competition mechanisms determine the levels of susceptible and resistant strains. Our model reproduces observed variation in the frequencies of resistance to penicillins and macrolides in Europe. In this region, ecological competition between susceptible and resistant strains explains more of the observed differences between countries than does immunological competition. Our results contribute to a better understanding of the processes driving the maintenance of antibiotic resistance in the absence of vaccination.

## 2. Materials and Methods

### 2.1. Epidemiological Framework

We model the transmission dynamics of pneumococcal strains according to the epidemiological framework by Obolski et al. [37]. We define pneumococcal strains by the combination of their serotype and antibiotic resistance profile to two distinct antibiotics. Similarly to the nomenclature used in previous studies, each strain’s genotype is conceptualised by a tuple {i,j}, where i determines the serotype and j the antibiotic resistance profile. We model two serotypes only, such that i∈{a,b}. For the entirety of our study, we define serotype a as a vaccine type (VT) and serotype b as a non-vaccine type (NVT). We also model two antibiotics, x and y, and consider only resistance to each independently, with j∈{00,01,10}, where j = 00 refers to sensitive strains, j = 01 to strains resistant only to antibiotic y, and j = 10 to strains resistant only to antibiotic x. For instance, a strain defined by i = a, j = 00 is of VT serotype a and is sensitive to both antibiotics; while a strain i = b, j = 01 is of NVT serotype b, sensitive to antibiotic x and resistant to antibiotic y. For simplicity, we do not consider double resistance, i.e., j = 11 in our model. This is consistent with relatively low estimates of double resistance to penicillin and macrolides identified in Europe in pneumococci before the pneumococcal vaccine introduction [38,39].

We keep the terminology of the original model formulation regarding the epidemiological parameters: γ is serotype-specific immunity; ψ is the probability that a carried susceptible strain (j = 00) will suppress host co-colonization by a resistant strain (j = 01 or j = 10) due to the fitness cost of antibiotic resistance (details below); 1/μ is the host life-span; 1/σ is the host carriage duration; R0 is the basic reproduction number and β is the transmission rate.

Individuals are born naive to all strains, with the size of the host population kept constant (deaths being replaced by newborns). Colonization results in carriage for an average duration of 1/σ days, from which recovery (clearance) may lead to complete serotype immunity if γ = 1 or partial life-long immunity if γ<1. Thus, if γ<1, hosts can be re-colonized by the same serotype throughout their lifetime, with probability 1−γ. Co-colonization of up to two strains is allowed, unless γ = 1 and strains belong to the same serotype. Within-host strain competition interferes with co-colonization if ψ>0, which captures the degree to which intrinsic fitness differences (e.g., growth rates) between resistant and sensitive strains may allow a currently carried sensitive strain (j = 00) to suppress co-colonization by a resistant strain (j = 01,j = 10). We emphasize that ψ models a form of competition between bacterial strains that is not mediated by immunity—from now onwards referred to ecological competition. The modelled processes related to an individual’s colonization and epidemiological states are summarised in Figure 1A. The full model description is provided in the Appendix A.

In this study we use a stochastic, individual-based model (IBM) implementation [40] of the original epidemiological framework [37]. We implement model events by sampling from a Poisson distribution with mean equal to the deterministic expectation of state transitions: e.g., at each time step, the number of deaths is obtained from a Poisson distribution with mean μ×N, where N is the total population size, after which individuals are randomly selected for a death event. The time unit is a day.

The original framework is expanded by including host-population structure and host movement within a meta-population formulation as in our previously published IBMs [41,42,43]. Although we model a closed population, we allow for random introductions of any strain under a fixed frequency 1/η; i.e., with probability η per time step a random host and strain are selected and colonization is attempted (which depends on the host’s epidemiological state as in Figure 1A,B). In summary, population structure is included by subdividing individual hosts into a spatially organised set of L×L communities of population size Nc, forming a squared lattice of L columns and L rows. Individuals are assumed to mix homogeneously within each community. Local movement is considered, such that the force of infection of a strain in a particular community is dependent on its carriage levels within that community and in the neighbouring communities. Neighbouring communities are defined as the ones directly up, down, left, and right. Long distance host mobility is also modelled and assumed to take place randomly across the meta-population. Host movement is modelled by allowing individuals to transmit to individuals at a different community with probability ω by selecting the source and target communities and individuals randomly; that is, every time step a Poisson distributed number of individuals, with mean $\omega \times y_{i,j}$, will attempt to transmit strain i,j to a random individual in the meta-population. The modelled processes related to population structure are summarised in Figure 1C. The model parametrization is given in the Model parameterization section of the Appendix A.

### 2.2. Antibiotic Resistance and Consumption Data

Data source used was the European Centre for Disease Prevention and Control (ECDC) from the surveillance atlas (antimicrobial resistance of invasive isolates reported to the EARS-NET) for the year 2005, vastly representative of a pre-PCV era across EU/EEA European countries (available at [44]). The year 2005 is the earliest made available by the ECDC. The data variables used were (i) the percent of samples sensitive to penicillins and macrolides (independently), and (ii) consumption defined by daily doses per 1000 people per day. We considered solely countries which had a number of samples larger than 50, which included 19 countries. Resistance level was aggregated across countries (EU) using a logistic regression model with random-effect intercepts for each country. Raw data on resistance per country and aggregated EU is presented in Appendix A; for consumption in Appendix A.

### 2.3. Approximate Bayesian Computation

To compare the model output to real-world estimates on the prevalence of antibiotic resistance, we employed approximate Bayesian computation (ABC). This method allowed us to approximate the posterior density of simulation parameters related to competition (ψ, γ), given priors on the parameters and data, when there is no explicit likelihood function [45], as is the case for our IBM framework. The priors used for the parameters (ψ, γ) were both uniformly distributed in [0.1, 0.9]. These priors were chosen for two reasons: First, we are not aware of any specific prior knowledge of these values - except that they are probably not close to zero or one. Second, the model run time is non-negligible, and is substantially increased when either of the competition parameters is close to zero or one. Hence, constrained by the relatively long computation time the model required, we sampled the prior a total of 6741 times (each with a unique combination of ψ and γ). We used three data points as “targets” for the ABC: (i) the observed proportion of resistance to penicillins or macrolides aggregated at the EU level; (ii) the observed relative proportion of NVT in some EU-related epidemiological contexts (Appendix A); (iii) the proportion of carriers with co-infection. The ABC distance function employed to our model was the Euclidean distance from the simulation results to targets (i–iii). The threshold ϵ for acceptance under the ABC framework was set to either 0.025 or 0.05, so that the closest 2.5 or 5% of simulations to the targets (i–iii) would not be rejected. Further details of the implementation of the ABC approach are given in the section Approximate Bayesian computation of the Appendix A.

## 3. Results

### 3.1. Pneumococcal Strain Dynamics

We first assessed average model behaviour with regards to the levels of ecological and immunological competition, for which the parameters ψ and γ were marginally varied across the ranges [0.1, 0.9]. We focused on average model behaviour under the default parameter set (see Model parameterization in the Appendix A), but alternative sets including examples of temporal dynamics are presented in Appendix A.

When immunological competition γ was high and above γ≈0.6, independently of the level of ecological competition ψ, model dynamics were generally incompatible with observed pneumococcal strain dynamics. In particular, carriage levels of VT and NVT serotypes tended to be similar (Figure 2A), there was no host co-infection (Figure 2B), and strains resistant to antibiotics dominated the population (Figure 2C). Such epidemiological dynamics are incompatible with observed pneumococcal dynamics, which ubiquitously include high co-infection [46,47], reinfection [48,49,50], and co-circulation of NVT and VT serotypes with dominance of VT [10,47,48,51,52]. This dynamic behaviour is commonly observed in multi-locus, multi-strain pathogen modelling frameworks such as the one used here, in which high immunological competition has been shown to universally lead to the emergence of discrete strain structures (e.g., [28]).

In contrast, when immunological competition γ was low to intermediate, the model better replicated known properties of pneumococcal strain dynamics, resulting in dominance of VT serotypes and strains sensitive to antibiotics, as well as allowing for host coinfection (Figure 2A–C). Generally, for low-intermediate γ (i.e., 0.35–0.55), the carriage level of NVT serotypes was extremely low (Figure 2A); there was no co-infection (Figure 2B); while strains resistant to antibiotics did not dominate the population, their prevalence generally decreased for higher ecological competition ψ (Figure 2C). In contrast, when γ was lower (i.e., <0.35), carriage levels of NVT serotypes increased with ecological competition (Figure 2A); host coinfection was possible with varying levels of ecological competition (Figure 2B); both coinfection and sensitivity to antibiotics decreased with increasing ecological competition (Figure 2B,C). These general behaviours across the competition parameter space (ψ, γ) did not change substantially in sensitivity analyses with different host-population structure and mobility values (Appendix A). This suggests that variation of demographic parameters, which often exists between different regions, does not play a significant role in changing the levels of co-existence between susceptible and resistant strains in our modelling framework.

Resorting to data representative of the pre-vaccination era in Europe (from ECDC, see the Antibiotic resistance and consumption data section), we explored under what ecological and immunological competition levels the model better approximated observed levels and variation of resistance to macrolides and penicillins (independently). According to ECDC data, aggregated resistance across EU countries was approximately 18.1% (95% CI 12.19–26.51%) for macrolides and 8.1% (95% CI 5.02–12.95%) for penicillins (Appendix A). While a wide range of γ and ψ values appeared compatible with these levels of resistance (Figure 2C), some of those combinations would nonetheless result in widely different levels of the NVT versus VT prevalence (Figure 2A) and levels of co-infection (Figure 2B). Using an ABC algorithm applied to a large number of numerical simulations exploring the model’s parameter space, we next set out to identify which combinations of γ and ψ would reproduce the observed levels of resistance while respecting prior knowledge on the relative proportions of NVT versus VT and host co-infection (see Materials and Methods section for details).

### 3.2. Strain Competition as Determinants of Antibiotic Resistance

As illustrated in Figure 3A,B, reproduction of pre-vaccination resistance levels, while respecting priors related to observed NVT versus VT prevalence and strain co-infection, identified particular competition parameter regions. The latter are characterized by a combination of low immunological (γ<0.25) and low to intermediate ecological competition (ψ>0.2, ψ<0.7), which are compatible with observations of recurrent pneumococcal infection [48,49,50], co-circulation of NVT and VT serotypes [10,47,48,51,52] as well as frequent co-infection [46,47].

For penicillins, a restricted range for immunological competition γ (0.12–0.18) and wide range of ecological competition ψ (0.3–0.7) were compatible with observed resistance levels (Figure 3C). In contrast, resistance to macrolides was compatible with the model output under a wider range of γ (0.1–0.25) and stricter and lower range of ψ (0.22–0.38). The intersection of the two compatible parameter regions for macroIides and penicillins was for γ≈0.175 and ψ≈0.375. Doubling the percentage of accepted simulations under the ABC framework (ϵ = 5%) or the target co-infection level (from 0.2 to 0.3) had no significant impact on the identified parameter regions (Appendix A). In contrast, accepting higher margins of error (e.g., ϵ = 10%) resulted in the ABC identifying substantially wider regions of the parameter space as compatible, thus making it difficult to infer relationships between the model parameterization and observed resistance levels.

We further examined the associations between the competition parameters and resistance to penicillins and macrolides, performing an ABC which utilized each EU country’s antibiotic resistance frequencies to each of the two mentioned antibiotic classes (see Appendix A for method details). For either of the antibiotic classes, ecological competition ψ explained a higher percent of the variance of resistance than immunological competition γ in a simple linear model (penicillins, $R^{2}$ of 0.9 versus 0.001; macrolides, $R^{2}$ of 0.71 versus 0.52, respectively). Furthermore, for either of the antibiotic classes, a likelihood ratio test concluded that a bivariate model including both γ and ψ did not have a significantly better goodness of fit over a model containing ψ alone. Hence, in our model, variations in ecological competition could better explain differences of antibiotic resistance frequencies in the pre-vaccination era.

## 4. Conclusions

How do ecological and evolutionary forces drive the coexistence of pathogen strains susceptible and resistant to treatments or prophylaxis? This question remains a conundrum of high public health concern. Examples of such coexistence include some of the most relevant human pathogens: antiviral resistance of the immunodeficiency virus (HIV) [53,54], antimalarial resistance to malaria parasites [55], drug and vaccine escape by *Mycobacterium tuberculosis* [56], and antimicrobial resistance in *Streptococcus pneumoniae* [19,33,57,58].

Specifically, the understanding of what allows resistant bacteria to persist in human populations with different epidemiological, demographic, antibiotic consumption and epidemiological control conditions remains elusive. Antibiotic consumption is considered a major driver of resistance at the population level for both commensal and pathogenic bacteria [57,59,60]. However, while antibiotic consumption undoubtedly selects for antibiotic resistance, a major fraction of the observed spatio-temporal variation in resistance prevalence remains to be explained. Other population characteristics, e.g., gross domestic product (GDP) per capita or public health-care spending, have also been observed to affect variation in resistance levels [61]. Indeed, assuming no substantial time lag between consumption and prevalence of resistance [62], the ECDC data here explored shows that country-specific resistance is only mildly correlated with antibiotic consumption across countries, with consumption explaining 0.14 ($R^{2}$) of the variation for macrolide resistance (p = 0.06) and 0.38 of penicillin resistance (p = 0.002) (Appendix A).

Mathematical models have the potential to explore possible mechanistic drivers of resistance by explicitly considering factors modulating selective pressures. In this context, most pneumococcal modelling frameworks commonly overlook within-host interactions between phenotypically divergent strains (although for exceptions see [21,63]). The concept of “no coexistence for free” is also key for interpretation of modelling output. That is, candidate models cannot include intrinsic mechanisms allowing for stable coexistence of indistinguishable strains. Only under the assumption of such neutral models is it possible to directly discern the effect of the explicitly modelled factors in defining the existence of a stable state characterised by coexistence [64]. Indeed, the model presented here does not offer coexistence for free, since modelling indistinguishable strains (γ = 0, ψ = 0) dictates dynamics subjected only to drift (stochasticity). In this study, we expand a modelling framework previously used to explore within-host competition between pneumococcal strains as a driver of post-vaccination increases in antibiotic resistance [37]. Informed by available data for European countries, we expand the framework to a pre-vaccination era. We explore how the combination of within-host ecological and immunological competition may drive observed population-level coexistence between resistant and susceptible pneumococcal strains. Other possible drivers of coexistence (e.g., age-dependent assortative mixing [60]) were not explored in this study, although we have shown that high-level heterogeneities such as population structure and host mobility do not significantly affect model output (Appendix A). Furthermore, our study did not consider double resistance to penicillins and macrolides. Unfortunately, the data we had access to did not contain the rates of double resistance to these antibiotics in the studied period. As such, we only model the marginal distributions of resistance to penicillins and macrolides. Although other data suggest that double resistance to macrolides and penicillins in this study’s setting are not high in pneumococci, they might not be negligible [38,39]. Future endeavors, informed by higher resolution epidemiological and molecular data on the interactions between the two resistance types, should also consider double resistance.

Our main results demonstrate the complex relationship between the frequency of NVT and VT pneumococci, co-infection, antibiotic resistance, and ecological and immunological competition factors. Based on data from European countries, our model indicates that pre-vaccination era antibiotic resistance levels are negatively associated with ecological competition. In our model, high ecological competition decreases antibiotic resistance levels, as this parameter governs the co-colonization competence of antibiotic resistant strains against susceptible strains, due to possible fitness costs incurred by acquisition of resistance [65]. The variation of ecological competition levels between susceptible and resistant strains across countries can be due to differences in both the bacterial and human populations. Different strains, with different genetic loads and, hence, resistance fitness costs, can expand across countries due to chance and founder effects [66,67]. Additionally, differences in human population density and demography, environmental conditions, per capita hospital beds, and other factors can substantially affect ecological competition and its influence on co-existence of different strains [68,69,70].

In contrast to the identified role of ecological competition in our model, we do not find that immunological competition significantly contributes to explaining the variation in observed resistance levels across European countries. In reality, immunological competition between strains is determined by a complex, largely unknown network of antigenic relationships between pneumococcal strains; with capsular polysaccharides believed to be the major determinant of strain-specific and cross-strain immunological responses [71]. As such, immunological competition levels would have to substantially differ between countries if they were a factor affecting rates of co-existence between resistant and susceptible strains. However, there is a lack of support for such substantial local differences in the antigenic determinants and overall prevalence levels of circulating antigenic groups [72,73]. Hence, the results of our model are compatible with the notion that population-level antigenic relationships and diversity are not dissimilar enough across countries to justify immunological competition as a major determinant of the co-existence of resistant and susceptible strains in the pre-vaccination era.

To conclude, we present a general yet explicit framework to model pneumococci, considering mechanisms of both between-host transmission and within-host competition as determinants of the pathogen’s population dynamics. In a series of simulated scenarios, we demonstrate that ecological competition during co-colonization between resistant and susceptible pneumococcal strains can explain much of the variation of resistance frequencies observed at the country level in the pre-vaccination era. Our results demonstrate the framework’s general potential to explore the drives of pneumococcal resistance dynamics. Elucidating the role of underlying competition-related components of pneumococcal dynamics improves the understanding of the mechanistic drivers for the emergence and maintenance of antibiotic resistance. Consequently, frameworks such as this can be used to project the influence of changes in vaccination or antibiotic treatment strategies on antibiotic resistance.

## Figures and Tables

**Figure 1 vaccines-09-00265-f001:**
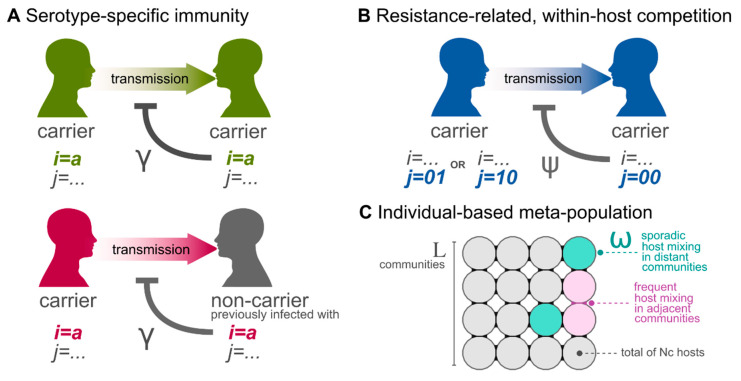
Model transmission assumptions and host-population structure. (**a**) Serotype-specific immunity as modelled by the parameter γ. Immunity can hinder the acquisition of a strain of a particular serotype (e.g., i = a) if (top) the receiving host is currently carrying a strain of that serotype (i = a), or (bottom) the receiving host is clear of infection (carriage) but has previously carried a strain of that serotype (i = a). (**b**) Resistance-related, within-host competition as modelled by the parameter ψ. Competition from an already established sensitive strain (j = 00) can hinder the acquisition of a strain carrying resistance to antimicrobials (j = 01 or j = 10). (**a**,**b**) The effects of immunity and competition are not mutually exclusive, and can take place at the same time if conditions shown in subplot a (top) and subplot b are met. (**c**) Schematic of the meta-population used, with L×L communities of size Nc hosts, with frequent mixing between adjacent communities, and sporadic events between distant communities with probability modelled by the parameter ω.

**Figure 2 vaccines-09-00265-f002:**
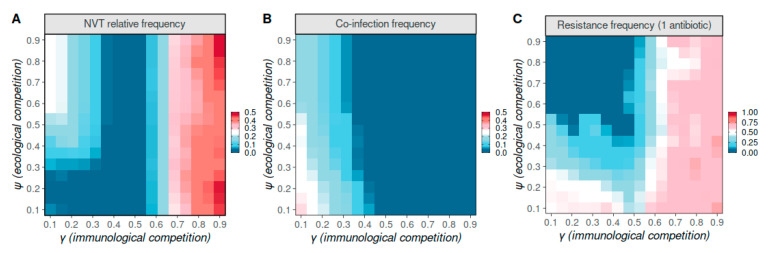
Model strain dynamics under variations to ecological and immunological competition. (**A**) Relative frequency (ratio) of total number of individuals carrying NVT versus carrying any type (NVT + VT). (**B**) Proportion of infected hosts carrying more than one strain (co-infection). (**C**) Relative frequency (ratio) of total number of individuals carrying resistant strains to one antibiotic (VT *j* = 10, VT *j* = 01, NVT *j* = 10, NVT *j* = 01) and carrying any strain. (**A–C**) All model parameters as in default parameter set except ψ, γ, varied in the y and x axis, respectively. Results presented are the mean over the last 5 years of a simulation, for particular combinations of ψ, γ.

**Figure 3 vaccines-09-00265-f003:**
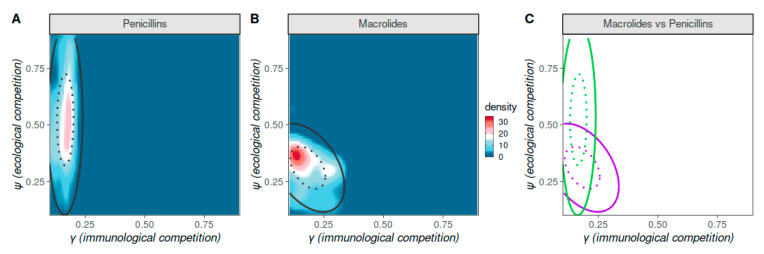
Competition parameter space compatible with observed, pre-vaccination resistance levels. Approximate Bayesian Computation (ABC) output when varying immunological (γ) and ecological (ψ) competition, attempting to reproduce observed levels of resistance to penicillins (**A**) and macrolides (**B**) in the European region (see Data section for details). The colour scale is the density in ABC posterior estimate when applied to the aggregated European region resistance levels. Ellipses mark the 50 (dotted) and 95 (full) percentiles in ABC output. ABC priors and targets as detailed in the Methods section, the number of simulations was 6741, ϵ was set to 2.5%. (**C**) Ellipses marking the percentiles in panels A (green) and B (purple).

## Data Availability

All data are in the public domain.

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
