# Peer review of "Pneumococcal Competition Modulates Antibiotic Resistance in the Pre-Vaccination Era: A Modelling Study"

_vaccines, 2021, doi:10.3390/vaccines9030265_

Round 1

Reviewer 1 Report

The manuscript of Lourenco et al. about a modelling study on pneumococcal competition has clearly been improved during revision.

Unfortunately, there is still a lack of data on double resistant strains (j = 11), but without access to real data no sincere calculation/modelling can be done.

Developing algorithms or predictable assumptions on fitness costs of resistance might be an interesting topic for the future of this project.

It would be also exciting to see the influence of age-dependent assortative mixing.

Author Response

We completely agree with the reviewer. These excellent points for future work are mentioned in the discussion. 

Other possible drivers of coexistence (e.g. age-dependent assortative mixing [69]) were not explored in this study, although we have shown that high-level heterogeneities such as population structure and host mobility do not significantly affect model output (Supplementary Figures S6-S7). Furthermore, our study did not consider double resistance to penicillins and macrolides. Unfortunately, the data we had access to did not contain the rates of double resistance to these antibiotics in the studied period. As such, we only model the marginal distributions of resistance to penicillins and macrolides. Although other data suggest that double resistance to macrolides and penicillins in this study’s setting are not high in pneumococci, they might not be negligible [47,48]. Future endeavors, informed by higher resolution epidemiological and molecular data on the interactions between the two resistance types, should also consider double resistance.”

Reviewer 2 Report

Much improved

Author Response

Thank you!

This manuscript is a resubmission of an earlier submission. The following is a list of the peer review reports and author responses from that submission.

Round 1

Reviewer 1 Report

Title: “Within-host competition…” is unclear. Do you actually mean something like ‘serotype replacement ‘ . Your publication would have greater impact if the title was clarified. Moreover, it needs to state that it is a modelling study.

Introduction is very long and rambling. The abstract and the last paragraph of the introduction leaves me wondering.

The abstract needs to state the data source and provide some information on the method (Bayesian modelling?). Some results in the abstract would be nice too! Even if only something like “Our main results demonstrate the complex relationship between the frequency of NVT and VT pneumococci, co-infection, antibiotic resistance, and ecological and immunological competition factors.”

Reviewer 2 Report

The manuscript of Lourenco et al. describes an interesting approach to mathematically model pneumococcal resistance rates based on ecological and immunological competition.

Yet, the paper has some shortcomings, as not all assumptions are really balanced on current knowledge.

For instance, the model defines strains either as susceptible to both antibiotic classes (e.g. penicillins or macrolides) or as resistant to only one of the classes (01, 10). A double-resistant situation is not modelled (line 100-105). In high resistant countries penicillin and macrolides resistance rates are up to 50% or higher (e.g. in Russia) and double-resistant strains (j = 11) would be most likely there.

The assumption is based on the pre-vaccine era, which probably existed in the eighties of the last century, but does not reflect a realistic scenario of today. A projection for j=11 should be included in the model.

The mathematical model is based on the assumption that immunological competition (immunity, strain type) and ecological competition (resistance) are independent variables. Zhao et al. has demonstrated in 2017 that resistance rates are not equally distributed. They showed high penicillin resistance rates e.g. for capsule types 19A, 19F or 14, but not for other serotypes. The selection pressure (immunological competition) is influenced by regional distributions and depends on the type of vaccine used (PCV7, PCV13 or PPSV23).

The conclusion that population structure would not affect model output sounds unlikely (l. 332-335). Life expectancy is quite different in Europe (e.g. 83 years in Italy vs. 71 in Moldavia), which results in a much higher ratio of younger population in the latter. Colonization levels of pneumococcus are age-dependent, with a span from >40% in under-fives to less than 2% above 25 years of age. Most pneumococcal infections are found in very young children or in adults above 60.

Overall the framework cannot yet explain the influence of vaccination on the development of antibiotic resistance in pneumococcus.

As long as clear predictions could not be made by this complex mathematical model the topic might be on the fringes of the reader’s interest and the scope of the journal.

Reviewer 3 Report

Introduction, results and conclusions well presented. Figures designed to the point.